# The Association of Social Support and Loneliness with Symptoms of Depression, Anxiety, and Posttraumatic Stress during the COVID-19 Pandemic: A Meta-Analysis

**DOI:** 10.3390/ijerph20042765

**Published:** 2023-02-04

**Authors:** Aina Gabarrell-Pascuet, Helena García-Mieres, Iago Giné-Vázquez, Maria Victoria Moneta, Ai Koyanagi, Josep Maria Haro, Joan Domènech-Abella

**Affiliations:** 1Epidemiology of Mental Health Disorders and Ageing Research Group, Sant Joan de Déu Research Institute, 08950 Esplugues de Llobregat, Spain; 2Research, Teaching, and Innovation Unit, Parc Sanitari Sant Joan de Déu, 08830 Sant Boi de Llobregat, Spain; 3Centro de Investigación Biomédica en Red de Salud Mental (CIBERSAM), Instituto de Salud Carlos III, 28029 Madrid, Spain; 4Department of Medicine, Universitat de Barcelona, 08007 Barcelona, Spain; 5Health Services Research Unit, Hospital del Mar Medical Research Institute (IMIM), 08003 Barcelona, Spain; 6Centro de Investigación Biomédica en Red de Epidemiología y Salud Pública (CIBERESP), Instituto de Salud Carlos III, 28029 Madrid, Spain; 7Institució Catalana de Recerca i Estudis Avançats (ICREA), 08010 Barcelona, Spain

**Keywords:** depressive symptoms, anxiety symptoms, posttraumatic stress symptoms, loneliness, social support, COVID-19

## Abstract

Background: Research suggests that changes in social support and loneliness have affected mental disorder symptoms during the COVID-19 pandemic. However, there are a lack of studies comparing the robustness of these associations. Aims: The aims were to estimate the strength of the associations of loneliness and social support with symptoms of depression, anxiety, and posttraumatic stress during the COVID-19 pandemic (2020–2022) in the general population. Method: The method entailed a systematic review and random-effects meta-analysis of quantitative studies. Results: Seventy-three studies were included in the meta-analysis. The pooled correlations of the effect size of the association of loneliness with symptoms of depression, anxiety, and posttraumatic stress were 0.49, 0.40, and 0.38, respectively. The corresponding figures for social support were 0.29, 0.19, and 0.18, respectively. Subgroup analyses revealed that the strength of some associations could be influenced by the sociodemographic characteristics of the study samples, such as age, gender, region, and COVID-19 stringency index, and by methodological moderators, such as sample size, collection date, methodological quality, and the measurement scales. Conclusions: Social support had a weak association with mental disorder symptoms during the COVID-19 pandemic while the association with loneliness was moderate. Strategies to address loneliness could be highly effective in reducing the impact of the pandemic on social relationships and mental health.

## 1. Introduction

Since the appearance of the coronavirus disease (COVID-19) in December 2019, one of the greatest concerns has been its effects on the general population’s mental health in both the short- and long-term. For example, the implementation of public health and social measures during the pandemic could have had a negative impact on social relationships [1,2], which in turn could have resulted in an adverse impact on mental health outcomes [3,4]. Indeed, the current evidence concerning the impact of the pandemic on the prevalence of mental disorders and their symptoms shows a significant increase in the general population [5]. Although the available studies consistently report an increasing trend, the use of different measures to assess mental disorders or their symptoms makes comparison between studies difficult, with a wide variation being reported. Specifically, the reported prevalence of depression ranges from 16% to 34%, anxiety from 15% to 38% [1,6,7,8], and post-traumatic stress disorder (PTSD) from 18% to 33% [9,10].

The effect of social relationships on mental health has been widely documented. Researchers have distinguished between subjective and objective aspects of social relationships, which often interact with each other as moderators [11] or mediators [12] impacting mental health. Objective factors refer to the characteristics of a social network described through quantifiable measures such as the number of close contacts or social interactions, whereas subjective factors refer to how individuals feel regarding that social network [13,14]. Social support and loneliness are, respectively, widely used measures for operationalizing these two types of factors. Social support has been defined as the instrumental, informational, and emotional support provided by a social network that includes family, friends, and neighbours [15] while loneliness has been defined as the unpleasant feeling that occurs because of the difference between the desired and the available social relationships, both quantitatively and qualitatively [16].

According to previous research, loneliness and low social support are among the social determinants most closely related to mental health compared to socioeconomic, material, and behavioural factors [13,17]. These relationships mainly occur with loneliness as the origin [18]. Pre-pandemic and during the pandemic investigations indicate that low social support boosts the development of loneliness [19,20,21,22] and that the effect of social support on mental health is mediated by loneliness [23,24,25]. Both factors increase the odds of having symptoms of depression and anxiety [11,18]. Post-traumatic stress symptoms (PTSS), as already observed during and after the SARS pandemic of 2003, are likely to appear and increase in the long-term following the COVID-19 pandemic, particularly among the most vulnerable groups (e.g., COVID-19 patients and their close contacts, health care workers and other hospital staff, persons with a psychiatric illness history or with underlying health conditions, older people, individuals who reside in high COVID-19 prevalence areas, etc.) [26,27,28]. This increase could also be aggravated by the effects of loneliness and poor social support [4].

The COVID-19 pandemic has generated unprecedented situations and posed unique challenges globally, leading to a fast and constantly growing body of scientific evidence related to the topic. Although there is now an expanding literature about objective and subjective aspects of social relationships, it is still unclear which constructs (i.e., social support or loneliness) have a higher impact on mental health and how this impact differs from the symptoms of one mental disorder to another. Clarifying these aspects would provide relevant information for the design of psychosocial interventions aimed at improving the population’s mental health, which is particularly necessary in the COVID-19 pandemic context.

Thus, the aim of this study was to systematically review quantitative studies published from 2020 to 2022 exploring the associations of loneliness and/or social support with mental disorder symptoms (i.e., depression, anxiety, and post-traumatic stress) during the COVID-19 pandemic. In order to estimate the strength of the associations among these variables, we aimed to perform a statistical meta-analysis, so as to be able to objectively combine and analyse the results of the selected studies.

## 2. Methods

The review’s protocol was registered in PROSPERO, which is an international prospective register of systematic reviews with protocols related to COVID-19 (registration number: CRD42021260142). The methodology followed the recommendations published in the PRISMA statement [29].

### 2.1. Eligibility Criteria

Literature included in this review was limited to journal articles using general population-based cohort studies measuring the associations of social support or loneliness (i.e., independent variables [IV]) with symptoms of depression, anxiety, or posttraumatic stress (PTS) (i.e., dependent variables [DV]). All the main variables had to be measured quantitatively using validated scales.

The publication period was restricted to the first three years from the appearance of COVID-19 (i.e., from January 2020 (1 January 2020) to October 2022 (3 October 2022)). Observational studies, both cross-sectional and longitudinal with cross-sectional associations between the variables of interest, were included. Only publications in English and Spanish were included.

Studies on the general adult population were included in this review, excluding cohorts of specific populations that the pandemic may have affected differently (e.g., medical staff, caregivers, patients of specific diseases or those in a hospital setting, pregnant women, etc.). We also excluded studies focused on older adults (>60 years) or on children (<16 years) due to the differences regarding mental health outcomes in these specific age groups [30,31,32]. Moreover, eligibility was restricted to studies with a sample size of 450 participants or more to guarantee that the included articles had enough statistical power to provide substantial estimates of the general population [33,34].

Finally, regarding the measures of interest, studies in which the variables (at least one IV and one DV) were measured quantitatively were included. We ruled out those studies that did not (i) use a valid mode of ascertainment of the measures of interest (e.g., studies that assessed the main variables with a single-item measure or with a non-validated scale, including self-developed scales and adaptations of valid scales), (ii) report disjoint data for each variable (e.g., studies reporting overall mental health as the DV), or (iii) contain relevant and/or complete data for the associations between the variables of interest.

### 2.2. Article Search, Identification, and Selection

The PubMed, ScienceDirect, and Web of Science databases were searched for relevant studies due to their relevance to the review’s objectives and scope. Separate search strategies were developed for each database (Appendix A). Key search terms for mental health outcomes were ‘depression’, ‘anxiety’, ‘post-traumatic stress’, and ‘mental health’. For the social determinants, the terms included were ‘loneliness’, ‘social connectedness’, ‘social isolation’, ‘social network’, ‘social relationships’, and ‘social support’. For the COVID-19 pandemic, we searched for ‘COVID-19’, ‘lockdown’, ‘pandemic’, and ‘quarantine’.

Figure 1 is a flow diagram of the search and inclusion process. The literature search resulted in 6211 publications (Figure 1). For the study selection, Rayyan reference manager app was used. After removing duplicates, 5239 publications were screened based on their titles and abstracts and categorized as ‘include’, ‘maybe’, or ‘exclude’ by two independent reviewers (AG-P & JD-A) based on the eligibility criteria (researchers were blinded to each other’s decisions). Subsequently, decisions of the two reviewers were merged, yielding a percentage of agreement higher than 95%. Discrepancies between the reviewers were resolved by consensus. Finally, the two reviewers independently reviewed the full text articles in the ‘include’ and ‘maybe’ categories (*n* = 259). The 186 studies excluded at the full-text screening stage were tabulated alongside the reason for exclusion in accordance with best practice guidelines [29,35].

### 2.3. Data Extraction

A total of 73 studies were included in the present review and meta-analysis. One systematic reviewer (AG-P) extracted the data from the selected studies into a structured template and assessed their methodological quality. A second reviewer validated all the extracted data (JD-A). The following data were extracted (where available): study details (first author, publication date, article title, study design, country, collection date, main inclusion/exclusion criteria, sample size (N), type of population, aims of the study, and data collection methods), sample characteristics (age, sex), statistical methods, social relationships and mental health measurements, adjustment for covariates, estimates of associations, and key findings. In the case of longitudinal studies, as just few studies used this design and they used distinct follow-up periods, we just included their cross-sectional baseline data.

### 2.4. Quality Assessment

To evaluate the methodological quality of the studies, we used an adapted version of the Newcastle Ottawa Scale (NOS) [36] for cross-sectional studies, used in previous systematic reviews [37,38] (Appendix A). The NOS checklist has three sections that examine different characteristics of the studies (i.e., selection, comparability, and outcome). Some items are rated with one star and others with two. The total score for each section is what determines the quality of the studies (i.e., 1 = ‘poor’, 2 = ‘fair’, or 3 = ‘good’). Any discrepancies in terms of rating were resolved between the reviewers.

### 2.5. Meta-Analytical Method

#### 2.5.1. Calculating Effect Sizes

All the analyses were done using the *meta* package [39] for R software [40].

For the meta-analysis, we required the correlation values of the cross-sectional relationships of interest between continuous variables. When correlations were not available, we converted equivalent statistical measures (e.g, odds ratio (OR)) to correlations. Regarding OR, it was necessary to use a single cut-off point that allowed comparison of people feeling loneliness or having poor social support with the rest of the population. Priority was given to non-adjusted OR and, when not available, to OR adjusted for basic sociodemographic variables (e.g., age and sex), but not for other variables with a potentially mediating role. When the independent variable had more than one category (e.g., low/moderate/high social support), it was dichotomized, and the OR was calculated comparing those with low social support to the remaining sample. Finally, ORs were transformed to Pearson’s *r* according to the following conversions [41,42,43]:

Odds ratio to Cohen’s *d*: $d=LogOddsRatio\times \frac{\sqrt{3}}{\pi }$

Cohen’s d to Pearson’s *r*: $r=\frac{d}{\sqrt{d^{2}+4}}$

In cases where the independent variable was divided into different dimensions from the construct ‘loneliness’ or ‘social support’, the average overall correlation between the different dimensions was obtained [44]. If the study met the inclusion criteria and none of the aforementioned options were possible, the corresponding authors of the original articles were contacted to obtain the required unreported data. Correlations were reported as positive when the relationship was what was expected (i.e., lower social support or higher loneliness directly related to greater mental disorder symptoms).

#### 2.5.2. Statistical Analysis

We conducted a series of random-effects meta-analyses, according to the relationships reported for each study [45]. We used the random effects model due to the high heterogeneity across studies. We assessed statistical heterogeneity using restricted maximum likelihood as a heterogeneity variance estimator with the I^2^ statistic, which describes the percentage of total variation across studies that is due to heterogeneity rather than to chance, and the among-study variance τ^2^, which is the random effects variance of the true effect sizes [46].

#### 2.5.3. Sources of Heterogeneity

In order to assess the sources of heterogeneity identified in the meta-analysis, we performed subgroup analyses. We evaluated (i) sociodemographic moderator variables and (ii) methodological moderator variables.

Sociodemographic moderator variables were proportion of females in the sample, mean age of the sample, economic region where the study was conducted, and COVID-19 stringency index. The COVID-19 stringency index [47] is a composite score between 0 and 100 designed to compare countries’ policy responses to the pandemic, where higher values represent greater strictness of ‘lockdown policies’ (i.e., closure and containment measures). For each study, the COVID-19 stringency index was determined according to the study setting and the first day of data collection.

Methodological moderator variables were sample size, collection date, study methodological quality, and type of measure to assess the dependent and independent variables. To classify the scales to measure the main variables, we distinguished between the most commonly used measures (i.e., UCLA for loneliness [48], PHQ for depressive symptoms [49], GAD for anxiety symptoms [50], and MSPSS for social support [51]) and “Others”. In the case of the measures used to assess PTSS, due to the concern about flawed published work caused by measuring PTSD related to the pandemic without adequately considering PTSD criteria [52], we performed sub-group analyses distinguishing between those studies that assessed traumatic stress symptoms relative to the COVID-19 pandemic compliant with the DSM-5 criteria [53], using updated measures, and specifying the symptomatic timeframe and those that did not. All the studies that met these criteria also used the PTSD Checklist for DSM-5 (PCL-5) scale [54], so the subgroup was called “PCL5”; while the studies that did not meet any of the criteria were classified in the “Other” group.

The studies that did not have available data regarding a covariate were excluded when carrying out the subgroup analysis for that covariate.

## 3. Results

### 3.1. Study Characteristics

The meta-analysis included 73 studies reporting 137 effect sizes from a correlation of either total social support or loneliness with symptoms of depression, anxiety, or PTS (Table 1). The total number of participants involved in the analysis was 1,020,461 (466–746,217 participants), with a mean age of 33.23 (SD = 10.39, not reported in 5 studies), and with around 61.5% (SD = 12.0%, [39.9–87.8%]) of the sample being female (not reported in 2 studies). Most of the study participants were from the general population (62%, N = 45), one third were college students (33%, N = 24), and 4 studies (5%) used samples of the general population with an overrepresentation or inclusion of only young adults (18–35 years). Studies were conducted mainly in China (29%, N = 21), in European countries (29%, N = 21), and in the United States (14%, N = 10). Due to the isolation and social-distancing measures that characterized the initial stages of the COVID-19 pandemic, study data collection was mainly with non-probabilistic sampling techniques via online platforms, social media channels, and email.

### 3.2. Meta-Analysis

Through random-effects meta-analyses, the six relationships of interest were studied: loneliness-depressive symptoms, loneliness-anxiety symptoms, loneliness-PTSS, social support-depressive symptoms, social support-anxiety symptoms, and social support-PTSS. The effect sizes of the association between loneliness and the mental health outcomes are presented in Figure 2. The pooled effect size for the association of loneliness with symptoms of depression, anxiety, and PTS were r = 0.49, r = 0.40, and r = 0.38, respectively. The three pooled effects represent a medium effect [127] characterised by a large degree of heterogeneity (I^2^ = 99%, I^2^ = 99%, and I^2^ = 98%, respectively). The correlations of the association between social support and mental health outcomes are presented in Figure 3. The pooled effect size for the association between social support and symptoms of depression, anxiety, and PTS were r = 0.29, r = 0.19, and r = 0.18, respectively. The effect of social support on the studied mental health outcomes was smaller when compared to loneliness. The pooled effects were characterised by a large degree of heterogeneity (I^2^ = 98%, I^2^ = 99%, and I^2^ = 97%, respectively).

### 3.3. Moderator Analysis

The heterogeneous results were analysed with subgroup analyses. For each of the subgroups, the total effects and associated heterogeneity measures were calculated and the results are reported in Table 2 and Table 3.

Although, in many cases, the moderation effects could not be tested in the relationships with PTSS due to a low number of studies, in general we observed that the associations of social support and loneliness with mental disorder symptoms were stronger in samples with a lower proportion of females and COVID-19 stringency index, in those studies adequately using the PCL-5 (Blevins et al., 2015), and conducted in China, whereas those studies using the Patient Health Questionnaire (PHQ) [49] to measure depressive symptoms and the UCLA loneliness scale [48] to measure loneliness showed weaker associations. In the case of the relationship between loneliness and mental disorder symptoms, the associations were stronger when the interviews were carried between July and December 2020 in studies with high methodological quality and in those studies using the Generalized Anxiety Disorder (GAD) scale [50] to measure anxiety symptoms while in developing countries, these associations were weaker. Regarding the relationship between social support and mental disorder symptoms, the correlation values were lower in those studies carried in Europe and with an earlier collection date. In the case of the sample size, the results were discordant between the studied associations.

### 3.4. Publication Bias

Publication bias was assessed by constructing funnel plots (Appendix A) followed by Egger tests. The results indicated insignificant levels of publication bias for all relationships, except for the associations of social support and loneliness with anxiety (*p* < 0.05).

## 4. Discussion

This meta-analysis sought to explore the correlation of social support and loneliness with symptoms of depression, anxiety, and posttraumatic stress during the COVID-19 pandemic. The results show that social support had a weak association with mental disorder symptoms, whereas loneliness had a moderate association with symptoms of anxiety, posttraumatic stress, and, particularly, depression during the COVID-19 pandemic. If we compare the pooled correlations with guidelines for interpreting the magnitude of correlation coefficients [128], we observe the effect of the association between loneliness and mental health outcomes to be in the upper third distribution of correlation coefficients. In all cases, the results were characterized by a high level of heterogeneity.

In general, our results suggest that the effect sizes of the associations of social support and loneliness with symptoms of mental disorders are similar to pre-pandemic evidence. A previous meta-analysis reported almost the same effect size of loneliness on depression (r = 0.50) [129] as the one obtained in the present study, whereas another meta-analysis showed a weaker association between social support and depression (r = 0.17) [130]. A systematic review defined the association between loneliness and anxiety as moderate (r = 0.42), whereas the association between social support and anxiety was seen to be less clear [17]. Finally, the effect size of the association between social support and PTSS reported in the present meta-analysis was lower than that reported in a previous study with pre-pandemic data (r = 0.26) [131], which may be explained by the diversity of traumatic events considered. No reviews were found about the effect of loneliness on PTSS in the general population.

Therefore, the increase in mental health problems during the COVID-19 pandemic could be partially explained by an increase in the prevalence of loneliness and a decrease in social support [1,2] rather than by more robust associations between social relationships and mental health. The smaller effect size obtained in those associations where social support was the independent variable could be explained by (i) the fact that loneliness might mediate the relationship between social support and mental health [23,24] and (ii) the substantial overlap between these two constructs and the instruments used to measure them. Loneliness and social support are both strictly linked with an individual’s social system and are interconnected concepts that affect one another. They partially share some conceptual aspects but are distinguished by the theoretical interpretation and definition of the concepts, the individual’s experience, perceptions, and social exchanges, and its public connotations [132,133].

Most of the moderation effects detected in our analysis can be explained by previous literature. The stronger association of social support with mental disorder symptoms among younger individuals can be explained by the different relational needs that exist across age groups. Following the socioemotional selectivity theory [134], although social contact declines across adulthood, social goals change and the close and emotionally satisfying relationships prevail, and these may have remained more stable during the pandemic. In contrast, young people rely more on frequent and diverse social interactions, which might have been more greatly affected by social restrictions [134,135]. On the other hand, this pattern is not seen for the impact of loneliness, as subjective aspects of social relationships do not necessarily correspond with objective ones.

Our results are also in line with previous evidence suggesting that the beneficial effects of social support on mental health are stronger in the most deprived regions and neighbourhoods (i.e., with lower socioeconomic conditions and social capital and higher poverty), where inhabitants would be more likely to establish reciprocity networks with neighbours due to the absence of other resources [136,137,138].

In addition, the stronger associations seen in those studies with a higher proportion of males in the sample could be explained by the cultural differences in the socialization process of men and women, as men might be more vulnerable to the negative consequences of loneliness and low social support on mental health since they have fewer relational resources due to different socialization processes between genders. Socialization among men tends to lead to an emotional independence from general social support, with men relying on their partners for social and emotional support, whereas socialization among women tends to lead to a more complex social and emotional life [139].

The differences among variable measurement instruments may be partially explained by the scales having been designed to measure different types of symptoms, the use of different terminology, and variations in recall time frames [140]. In the case of COVID-19 related (post) traumatic stress symptoms, the associations were stronger when the DSM-5 criteria had been strictly followed; nevertheless, we should be cautious when interpreting these results, due to the low number of studies that fit in the “PCL5” category.

Finally, the moderation effect of the collection date and the COVID-19 stringency index, which is based on the public health and social measures imposed by the governments, reflect the changing course of the pandemic and its stages, which have affected social relationships and mental health differently.

### Strengths and Limitations

This is the first meta-analysis to focus on synthesizing correlational data of social support and loneliness with symptoms of depression, anxiety, and posttraumatic stress during the COVID-19 pandemic. A strength of the study is the exhaustive search of both published and unpublished data (i.e., multiple attempts to contact authors to obtain missing data) that it involved. However, the cross-sectional nature of the data from the included studies limited the possibility of examining causal relationships. It would have strengthened the meta-analysis to include prospective and longitudinal studies, but few studies used this design, and they used distinct follow-up periods, which impeded their inclusion. Second, the gathered data were based on self-reported questionnaires, which may have resulted in recall or reporting bias. In addition, self-reported measurements are related with the distorted perception of individuals with mental disorder symptoms and their mood state, which could have influenced some of the findings [141]. Third, although 22 out of the 73 included studies had a poor methodological quality, the exclusion of the poor quality studies did not have an impact on our overall results, as it can be seen in the subgroup analyses. Finally, the funnel plots revealed significant publication bias for the relationships with anxiety as the outcome. The asymmetry could be the result of publication and citation bias, as studies giving stronger results are more likely to be published and to be cited and, thus, are more likely to be included in meta-analyses [142,143]. However, no significant publication bias was detected in the remaining associations, although all of them reported a high level of heterogeneity (I^2^ > 95%). We explored methodological and theoretical factors moderating the correlation of the associations, but it is likely that other factors such as sociodemographic and socioeconomic characteristics of the study samples may have also contributed to the heterogeneity of the results. Therefore, we should cautiously interpret the findings of the present study, and future studies should try to identify further explanatory factors.

## 5. Conclusions

The current review shows that social support had a weak association, whereas loneliness had a moderate one, with mental disorders symptoms during the COVID-19 pandemic. Therefore, strategies focused on loneliness could be highly effective in reducing the impact of a pandemic on mental health. The synergy between objective aspects of social relationships, such as social support, and subjective aspects, such as loneliness, that configure the population’s mental health suggests that these interventions should be oriented both toward the individual and the community of social networks. These interventions directed towards people feeling loneliness should aim (i) to provide psychological assistance promoting changes in their social behaviour (i.e., targeting their maladaptive social perception and cognitive biases towards loneliness [144,145]) and (ii) to increase their chances of establishing satisfactory social contacts while considering the target population and the effects of moderator variables, such as gender, setting, and age.

## Figures and Tables

**Figure 1 ijerph-20-02765-f001:**
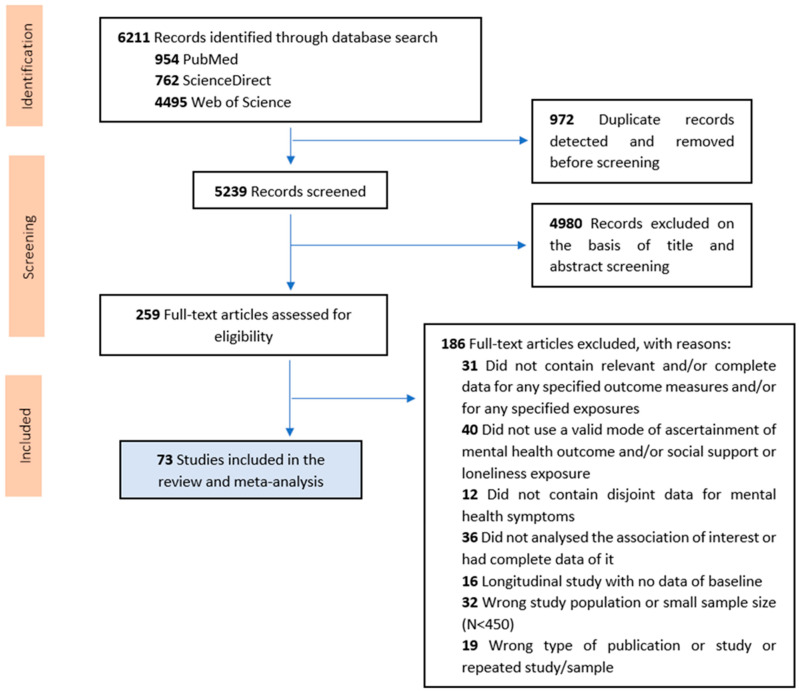
PRISMA flow diagram outlining results of the study selection process.

**Figure 2 ijerph-20-02765-f002:**
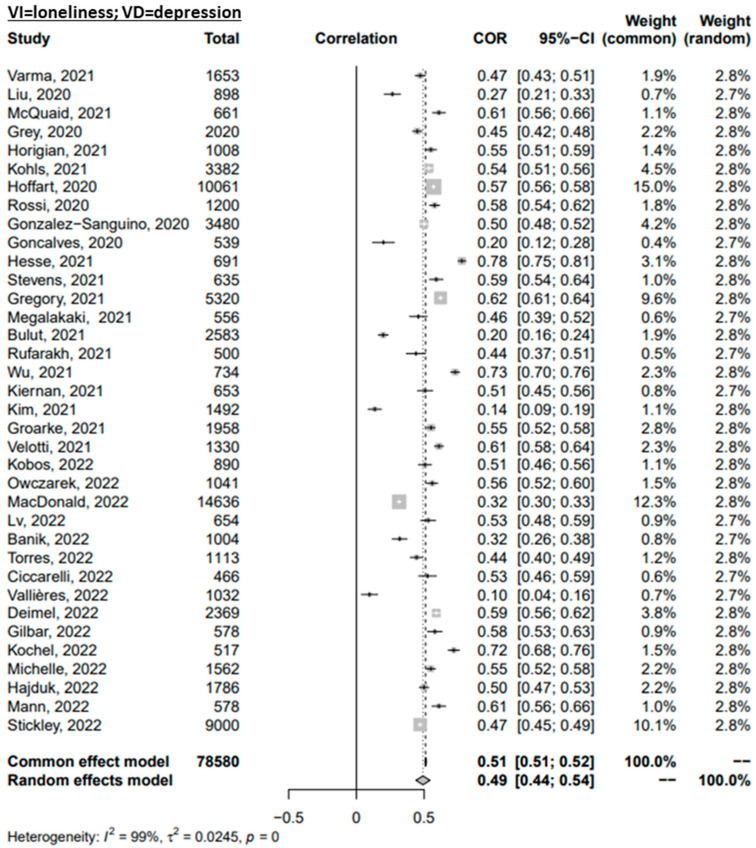
Forest plot of the Omnibus Test for the correlations of loneliness with mental disorder symptoms. References: Banik, 2022 [56]; Boursier, 2020 [59]; Boyraz, 2020 [60]; Bulut, 2021 [61]; Ciccarelli, 2022 [63]; Cordaro, 2021 [64]; Deimel, 2022 [65]; Gilbar, 2022 [70]; Gonçalves, 2020 [71]; González-Sanguino, 2020 [72]; Gregory, 2021 [73]; Grey, 2020 [74]; Groarke, 2021 [75]; Hajduk, 2022 [77]; Hesse, 2021 [78]; Hettich, 2022 [79]; Hoffart, 2020 [80]; Horigian, 2021 [81]; Kiernan, 2021 [87]; Kim, 2021 [88]; Kobos, 2022 [89]; Kochel, 2022 [90]; Kohls, 2021 [91]; Lim, 2022 [94]; Liu, 2020 [4]; Lv, 2022 [95]; MacDonald, 2022 [98]; Mann, 2022 [99]; McQuaid, 2021 [100]; Megalakaki, 2021 [101]; Orozco-Vargas, 2022 [104]; Owczarek, 2022 [106]; Rossi, 2020 [108]; Rufarakh, 2021 [109]; Stevens, 2021 [115]; Stickley, 2022 [116]; Torres, 2022 [119]; Vallières, 2022 [120]; Varma, 2021 [121]; Velotti, 2021 [122]; Wu, 2022 [84].

**Figure 3 ijerph-20-02765-f003:**
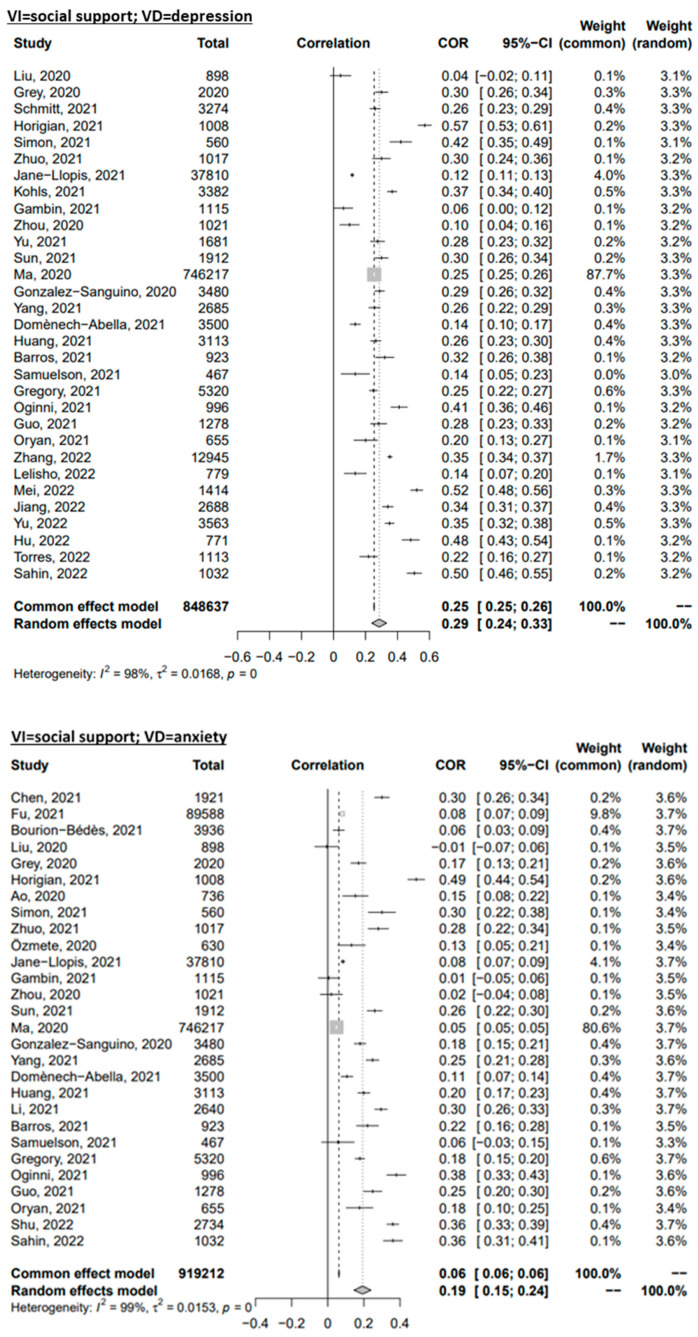
Forest plot of the Omnibus Test for the correlations of social support with mental disorder symptoms. References: Ao, 2020 [55]; Barros, 2021 [57]; Bourion-Bédès, 2021 [58]; Chen, 2021 [62]; Domènech-Abella, 2021 [66]; Fu, 2021 [67]; Gambin, 2021 [68]; Gan, 2022 [69]; González-Sanguino, 2020 [72]; Gregory, 2021 [73]; Grey, 2020 [74]; Guo, 2021 [76]; Horigian, 2021 [81]; Hu, 2022 [82]; Huang, 2021 [83]; Jané-Llopis, 2021 [85]; Jiang, 2022 [86]; Kohls, 2021 [91]; Lelisho, 2022 [92]; Li, 2021 [93]; Liu, 2020 [4]; Ma, 2020 [97]; Mei, 2022 [102]; Oginni, 2021 [103]; Oryan, 2021 [105]; Özmete, 2020 [107]; Sahin, 2022 [110]; Samuelson, 2021 [111]; Schmitt, 2021 [112]; Shu, 2022 [113]; Simon, 2021 [114]; Sun, 2021 [117]; Yu, 2022 [118]; Torres, 2022 [119]; Yang, 2021 [123]; Yu, 2021 [96]; Zhang, 2022 [124]; Zhou, 2020 [125]; Zhuo, 2021 [126].

**Table 1 ijerph-20-02765-t001:** Overview of included studies.

Authors	N	Mean (SD) Age [Age Range]	Gender (% Female)	Country	Measures of Loneliness and Social Support	Measures of Mental Health Symptomatology	Quality Rating	Reported Relationships
[55]	736	45 *	58.3	China	ss, SSRS	anx, STAI	Fair	ss-anx
[56]	1004	25.41 (7.80)	48.2	Bangladesh	lon, UCLA-3	anx, GAD-7 dep, PHQ-9	Good	lon-anx lon-dep
[57]	923	20.66 (4.27)	71.2	Portugal	ss, MSPSS	anx, DASS-21 dep, DASS-21	Poor	ss-anx ss-dep
[58]	3936	21.7 (4.00)	70.6	France	ss, MSPSS	anx, GAD-7	Fair	ss-anx
[59]	715	31.70 (10.81) [18–72]	71.5	Italy	lon, ILS-20	anx, DASS-21	Fair	lon-anx
[60]	747	41.26 (11.57) [22–76]	49.0	US	lon, UCLA-3	ptss, PCL-5	Good	lon-ptss
[61]	2583	22.84 (4.79)	65.5	Turkey	lon, UCLA-3	dep, CES-D-8	Poor	lon-dep
[62]	1921	29.28 (10.66) [16–68]	69.5	China	ss, MSPSS	anx, SAS	Fair	ss-anx
[63]	466	22.24 (2.68) [18–29]	45.5	Italy	lon, UCLA	anx, DASS-21 dep, DASS-21	Poor	lon-anx lon-dep
[64]	2101	47.80 (12.9)	87.8	US	lon, UCLA	anx, GAD-7	Good	lon-anx
[65]	2369	42.6 *	45.93	Germany	ss, OSSS-3 lon, De Jong	anx, GAD-7 dep, PHQ-9	Good	lon-anx lon-dep
[66]	3500	49.25 (15.64) [18–94]	51.5	Spain	ss, OSSS-3	anx, GAD-7; dep, PHQ-8	Good	ss-anx ss-dep
[67]	89,588	24 * [18–30]	56.3	China	ss, MSPSS	anx, GAD-7	Poor	ss-anx
[68]	1115	45 * [18–85]	50.5	Poland	ss, MOS-SSS	anx, GAD-7; dep, PHQ-9	Poor	ss-anx ss-dep
[69]	1390	30.7 * [14–67]	57.19	China	ss, MSPSS-6	ptss, PCL-5	Good	ss-ptss
[70]	578	45.2 (16.15)	57	Israel	lon, UCLA-3	anx, GAD-7 dep, PHQ-9	Good	lon-anx lon-dep
[71]	539	37.04 (12.91) [18–76]	75.7	Brazil	lon, UCLA	anx, GAD-7; dep, CES-D;	Fair	lon-anx lon-dep
[72]	3480	37.92 [18–80]	75.0	Spain	ss, MSPSS; lon, UCLA-3	anx, GAD-2; dep, PHQ-2; ptss, PCL-C-2	Poor	lon-anx lon-dep lon-ptss ss-anx ss-dep ss-ptss
[73]	5320	48.5 *	59.9	Canada	ss, MSPSS; lon, UCLA-3	anx, GAD-7 dep, PHQ-9	Good	lon-anx lon-dep ss-anx ss-dep
[74]	2020	24 *	50.0	Lebanon	ss, MSPSS; lon, UCLA-3	anx, GAD-7; dep, PHQ-9	Poor	lon-anx lon-dep ss-anx ss-dep
[75]	1958	37.01 (12.81) [18–87]	69.8	UK	lon, UCLA-3	dep, PHQ-9	Fair	lon-dep
[76]	1278	20.1 *	64.6	China	ss, PSSS	anx, DASS-21 dep, DASS-21	Fair	ss-anx ss-dep
[77]	1786	22.15 (3.53)	79.6	Slovakia	lon, UCLA-3	anx, GAD-7 dep, PHQ-9	Poor	lon-anx lon-dep
[78]	691	37.08 (10.85) [20–78]	43.6	US	lon, UCLA	dep, CES-D10	Good	lon-dep
[79]	2503	45.99 (17.77)	53.1	Germany	lon, UCLA-3	anx, HADS-6	Good	lon-anx
[80]	10,061	36.00 (13.5) [18–86]	78.2	Norway	lon, UCLA-8	anx, GAD-7; dep, PHQ-9	Good	lon-anx lon-dep
[81]	1008	28.09 (4.1) [18–35]	48.2	US	ss, SC-15; lon, UCLA	anx, GAD-7; dep, CES-D-10	Fair	lon-anx lon-dep ss-anx ss-dep
[82]	771	54	54	China	ss, MSPSS	anx, GAD-7 dep, CES-D	Good	ss-anx ss-dep
[83]	3113	20.83 (1.53)	71.4	China	ss, PSSS	anx, DASS-21 dep, DASS-21	Good	ss-anx ss-dep
[84]	734	20.35 (1.65)	46.9	China	lon, UCLA	anx, SAS dep, CESD20	Good	lon-anx lon-dep
[85]	37,810	45 *	74.1	Spain	ss, OSSS-3	anx, GAD-7; dep, PHQ-8	Good	ss-anx ss-dep
[86]	2688	20.49 [20,21]	NA	China	ss, SSRS	dep, SDS	Poor	ss-dep
[87]	653	40.3 *	84.4	Australia	lon, UCLA	anx, GAD-7 dep, PHQ-9	Poor	lon-anx lon-dep
[88]	1492	40.30 (11.8) [19–65]	50.1	South Korea	lon, UCLA-3	dep, PHQ-9	Fair	lon-dep
[89]	890	44.3 (16.1)	50.8	Poland	lon, R-UCLA	anx, HADS-M dep, HADS-M	Good	lon-anx lon-dep
[90]	517	19.52 (1.26)	57.4	US	lon, Context	anx, GAD-7 dep, CES-D	Poor	lon-anx lon-dep
[91]	3382	23.98 (4.66) [17–61]	70.2	Germany	ss, ESSI; lon, UCLA-3	dep, PHQ-9	Poor	lon-dep ss-dep
[92]	779	NA	61	Ethiopia	ss, OSSS-3	dep, CES-D	Poor	ss-dep
[93]	2640	20.66 [18–25]	68.8	China	ss, SSQ	anx, SAS	Poor	ss-anx
[94]	1562	48.8 [18–91]	84.2	Australia, UK, US	lon, UCLA	dep, PHQ-8	Good	lon-dep
[4]	898	24.5 [18–30]	81.3	US	ss, MSPSS; lon, UCLA-3	anx, GAD-7; dep, PHQ-8; ptss, PCL-C	Fair	lon-anx lon-dep lon-ptss ss-anx ss-dep ss-ptss
[95]	654	19.98 (1.80) [18–29]	50.31	China	lon, UCLA-8	dep, SDS	Fair	lon-dep
[96]	1681	20 *	64.8	China	ss, MSPSS	dep, CES-D	Good	ss-dep
[97]	746,217	20.20 *	55.6	China	ss, MSPSS	anx, GAD-7; dep, PHQ-9; ptss, IES-6	Good	ss-anx ss-dep ss-ptss
[98]	14,636	48 *	51.7	US	lon, UCLA-3	anx, PROMIS-4 dep, PHQ-8	Good	lon-anx lon-dep
[99]	578	39.22 (14.27) [18–79]	59.5	US	lon, De Jong-6	anx, DASS-21 dep, DASS-21	Good	lon-anx lon-dep
[100]	661	44 *	77.3	Canada	lon, UCLA-8	anx, GAD-7; dep, PHQ-9	Good	lon-anx lon-dep
[101]	556	30.06 (14.38) [18–87]	75.5	France	lon, UCLA-3	anx, GAD-7 dep, PHQ-9	Fair	lon-anx lon-dep
[102]	1414	NA	50.6	China	ss, SSRS	dep, SCL-90	Poor	ss-dep
[103]	996	29.00 (8.89)	48.1	Nigeria	ss, MSPSS	anx, HADS dep, HADS	Fair	ss-anx ss-dep
[104]	824	20.41 (1.29) [17–25]	55.09	Mexico	lon, De Jong	anx, BAI	Poor	lon-anx
[105]	655	38.6 * [18–86]	62.6	Israel	ss, MSPSS	anx, GAD-7 dep, PHQ-9	Fair	ss-anx ss-dep
[106]	1041	44.97 (15.76) [18–88]	51.5	Ireland	lon, UCLA-3	anx, GAD-7 dep, PHQ-9	Good	lon-anx lon-dep
[107]	630	39.20 [24–79]	73.0	Turkey	ss, MSPSS	anx, STAI	Poor	ss-anx
[108]	1200	39.33 (12.283) [18–81]	81.9	Italy	lon, UCLA	anx, SCL-90R; dep, SCL-90R	Good	lon-anx lon-dep
[109]	500	NA [18–40]	50.6	Pakistan	lon, UCLA	anx, DASS-21 dep, DASS-21	Poor	lon-anx lon-dep
[110]	1032	36.5 *	57	Turkey	ss, MSPSS	anx, DASS-42 dep, DASS-42	Good	ss-anx ss-dep
[111]	467	33.14 (12.96) [18–85]	63.6	US	ss, MSPSS	anx, GAD-7 dep, PHQ-9 ptss, PCL-5	Good	ss-anx ss-dep ss-ptss
[112]	3274	42.39 (13.41)	79.4	Brazil	ss, MOS-SSS	dep, PHQ-9	Poor	ss-dep
[113]	2734	20 [16–24]	48.24	China	ss, SSRS	anx, SAS	Poor	ss-anx
[114]	560	40.22 (11.60) [18–79]	74.0	Austria	ss, MSPSS	anx HADS; dep, HADS	Good	ss-anx ss-dep
[115]	635	43.52 (18.41)	48.5	Australia	lon, UCLA-3	dep, DASS-21	Good	lon-dep
[116]	9000	49.4 *	50.4	China	lon, UCLA-3	anx, GAD-7 dep, PHQ-9	Good	lon-anx lon-dep
[117]	1912	20.28 (2.10) [18–49]	69.8	China	ss, MSPSS	anx, GAD-7; dep, PHQ-9; ptss, IES	Fair	ss-anx ss-dep ss-ptss
[118]	3563	NA	68.57	China	ss, SSRS	dep, DBI-II	Poor	ss-dep
[119]	1113	21.45 (5.25) [18–100]	70.2	USA, Mexico, Ecuador, Spain, Chile	ss, MSPSS lon, UCLA	dep, PHQ-9 ptss, PCL-5	Fair	lon-dep lon-ptss ss-dep ss-ptss
[120]	1032	44.86 (15.74)	51.9	Ireland	lon, UCLA-3	anx, GAD-7 dep, PHQ-9	Good	lon-anx lon-dep
[121]	1653	42.90 (13.63)	69.7	63 countries	lon, UCLA-3	anx, STAI; dep, PHQ-9	Poor	lon-anx lon-dep
[122]	1330	NA	NA	Italy	lon, UCLA	anx, DASS-21 dep, DASS-21	Fair	lon-anx lon-dep
[123]	2685	27.00	39.9	China	ss, PSSS	anx, GAD-7	Fair	ss-anx ss-dep
[124]	12,945	21.5 * [17–26]	57.3	China	ss, MSPSS	dep, PHQ-9	Fair	ss-dep
[125]	1021	45.30 (16.46) [18–90]	52.3	US	ss, F-SozU K-6	anx, DASS-21 dep, DASS-21 ptss, PTGI	Good	ss-anx ss-dep ss-ptss
[126]	1017	20 *	53.3	China	ss, SSQ-6	anx, GAD-7; dep, PHQ-9	Good	ss-anx ss-dep

NOTE: N = frequency; NA = not available; * = mean age calculated from study data; lon = loneliness; ss = social support; anx = anxiety symptoms; dep = depressive symptoms; ptss = posttraumatic stress symptoms. Variables assessment measures: BAI = Beck Anxiety Inventory; BDI = Beck Depression Inventory; CES-D = Center for Epidemiologic Studies-Depression (8 or 20 items); Context = Loneliness in Context scale; DASS = Depression Anxiety Stress Scales; De Jong = De Jong Gierveld Loneliness scale (6 or 11 items); ESSI = Enriched Social Support Instrument; F-SozU K-6 = Social Support Questionnaire short form; GAD = Generalized Anxiety Disorder (7 items); HADS = Hospital Anxiety and Depression Scale; IES = Impact of Event Scale; ILS = Italian Loneliness Scale (20 items); MOS-SSS = Medical Outcomes Study—Social Support Survey; MSSPS = Multidimensional Scale of Perceived Social Support (6 or 12 items); OSSS = Oslo Social Support Scale; PCL = Posttraumatic Stress Disorder Checklist (‘-C’ based on DSM-IV or ‘-5′ based on DSM5); PHQ = Patient health questionnaire (2, 8, or 9 items); PROMIS = Patient-Reported Outcome Measurement Information System; PSSS = Perceived Social Support Scale; PTGI = Posttraumatic Growth Inventory; SAS = Self-rating anxiety scale; SC = Social Connectedness Scale (15 items); SCL-90 = Symptom Checklist 90; SDS = Self-Rating Depression Scale; SF-CiOQ = Short form of the changes in outlook questionnaire; SSQ = Social Support Questionnaire; SSRS = Social Support Rating Scale; STAI = State-Trait Anxiety Inventory; UCLA = University California–Los Angeles loneliness scale (3, 8, or 20 items).

**Table 2 ijerph-20-02765-t002:** Subgroup analysis for the associations between loneliness and mental disorder symptoms.

DV = Loneliness; IV = Depression						
Covariate	K	r	Lower CI	Upper CI	Heterogeneity	*p*-Value
Proportion of females (k = 35)						
<50%	8	0.57	0.47	0.67	98%	<0.001
50 to 70%	15	0.45	0.36	0.55	99%	
>70%	12	0.48	0.41	0.55	95%	
Age groups (k = 34)						
<30 years	12	0.48	0.39	0.57	98%	<0.001
30 to 40	8	0.53	0.42	0.64	98%	
>40 years	14	0.47	0.38	0.56	99%	
Region (k = 36)						
China	3	0.58	0.42	0.73	99%	<0.001
Europe	13	0.51	0.44	0.58	96%	
Developed	15	0.52	0.43	0.60	99%	
Developing	5	0.32	0.21	0.43	96%	
Stringency index (k = 29)						
<70	10	0.52	0.42	0.61	98%	<0.001
70–80	13	0.49	0.39	0.59	99%	
>80	6	0.48	0.41	0.56	93%	
Sample size (k = 36)						
<1000	15	0.54	0.46	0.62	97%	<0.001
1000–2000	12	0.45	0.35	0.55	98%	
>2000	9	0.47	0.38	0.56	99%	
Collection date (k = 31)						
January-June 2020	21	0.47	0.39	0.55	99%	<0.001
July-December 2020	8	0.54	0.48	0.60	97%	
2021–2022	2	0.38	0.26	0.51	91%	
Methodological quality (k = 36)						
Poor	10	0.48	0.41	0.56	98%	<0.001
Fair	9	0.42	0.31	0.53	98%	
Good	17	0.53	0.46	0.61	99%	
Depression measures (k = 36)						
PHQ	22	0.46	0.40	0.52	99%	<0.001
CES	6	0.53	0.32	0.74	99%	
Other	8	0.55	0.51	0.59	78%	
Loneliness measures (k = 36)						
UCLA	33	0.48	0.42	0.53	99%	<0.001
Other	3	0.64	0.56	0.72	93%	
**DV = Loneliness; IV = Anxiety**						
**Covariate**	**K**	**r**	**Lower CI**	**Upper CI**	**Heterogeneity**	***p*-Value**
Proportion of females (k = 30)						
<50%	6	0.45	0.29	0.61	100%	<0.001
50 to 70%	13	0.42	0.34	0.50	99%	
>70%	11	0.33	0.24	0.43	98%	
Age groups (k = 29)						
<30 years	9	0.38	0.31	0.44	92%	<0.001
30 to 40	7	0.34	0.21	0.46	97%	
>40 years	13	0.44	0.34	0.55	100%	
Region (k = 31)						
China	2	0.39	0.24	0.54	95%	<0.001
Europe	13	0.42	0.33	0.52	99%	
Developed	11	0.41	0.32	0.50	99%	
Developing	5	0.29	0.15	0.44	94%	
Stringency index (k = 26)						
<70	7	0.53	0.42	0.63	99%	<0.001
70–80	12	0.35	0.25	0.45	99%	
>80	7	0.34	0.24	0.44	96%	
Sample size (k = 31)						
<1000	14	0.37	0.29	0.45	94%	<0.001
1000–2000	8	0.39	0.30	0.49	96%	
>2000	9	0.44	0.30	0.57	100%	
Collection date (k = 31)						
January-June 2020	17	0.37	0.28	0.47	100%	<0.001
July-December 2020	7	0.48	0.44	0.41	88%	
2021–2022	1	0.42	0.37	0.47	-	
Methodological quality (k = 31)						
Poor	9	0.41	0.37	0.46	87%	<0.001
Fair	6	0.29	0.16	0.43	96%	
Good	16	0.42	0.34	0.51	100%	
Anxiety measures (k = 31)						
GAD	19	0.41	0.32	0.49	99%	<0.001
DASS	5	0.39	0.35	0.43	51%	
Other	7	0.37	0.28	0.46	99%	
Loneliness measures (k = 31)						
UCLA	26	0.37	0.32	0.43	99%	<0.001
Other	5	0.51	0.34	0.68	99%	
**DV = Loneliness; IV = Posttraumatic Stress**						
**Covariate**	**K**	**r**	**Lower CI**	**Upper CI**	**Heterogeneity**	***p*-Value**
Proportion of females (k = 4)						
<50%	1	0.51	0.46	0.56	-	<0.01
50 to 70%	0	-	-	-	-	
>70%	3	0.34	0.16	0.51	98%	
Age groups (k = 4)						
<30 years	2	0.37	0.09	0.65	98%	<0.01
30 to 40	1	0.27	0.24	0.30	-	
>40 years	1	0.51	0.46	0.56	-	
Region (k = 4)						
China	0	-	-	-	-	<0.01
Europe	1	0.27	0.24	0.30	-	
Developed	3	0.42	0.23	0.60	97%	
Developing	0	-	-	-	-	
Stringency index (k = 3)						
<70	0	-	-	-	-	-
70–80	3	0.34	0.16	0.51	97%	
>80	0	-	-	-	-	
Sample size (k = 4)						
<1000	2	0.37	0.09	0.65	98%	<0.01
1000–2000	1	0.51	0.47	0.56	-	
>2000	1	0.27	0.24	0.30	-	
Collection date (k = 4)						
January-June 2020	3	0.34	0.16	0.51	97%	<0.01
July-December 2020	0	-	-	-	-	
2021–2022	1	0.51	0.47	0.56	-	
Methodological quality (k = 4)						
Poor	1	0.27	0.24	0.30	-	<0.01
Fair	2	0.37	0.09	0.65	98%	
Good	1	0.51	0.46	0.56	-	
PTSS measures (k = 4)						
PCL5	1	0.51	0.46	0.56	-	<0.01
Other	3	0.34	0.16	0.51	98%	
Loneliness measures (k = 4)						
UCLA	4	0.38	0.23	0.53	98%	<0.01
Other	0	-	-	-	-	

NOTE: Some studies were excluded from the subgroup analysis due to missing values.

**Table 3 ijerph-20-02765-t003:** Subgroup analysis for the associations between social support and mental disorder symptoms.

DV = Social Support; IV = Depression						
Covariate	K	r	Lower CI	Upper CI	Heterogeneity	*p*-Value
Proportion of females (k = 30)						
<50%	4	0.38	0.25	0.52	98%	<0.001
50 to 70%	17	0.27	0.21	0.34	98%	
>70%	9	0.26	0.18	0.33	98%	
Age groups (k = 28)						
<30 years	17	0.31	0.26	<0.001	97%	<0.001
30 to 40	4	0.29	0.13	0.44	97%	
>40 years	7	0.19	0.10	0.28	97%	
Region (k = 31)						
China	12	0.33	0.28	<0.001	98%	<0.001
Europe	7	0.24	0.14	0.34	99%	
Developed	8	0.25	0.12	0.39	98%	
Developing	4	0.28	0.17	0.39	93%	
Stringency index (k = 27)						
<70	6	0.36	0.24	0.47	98%	<0.001
70–80	15	0.26	0.20	0.32	97%	
>80	6	0.21	0.13	0.30	97%	
Sample size (k = 31)						
<1000	8	0.27	0.16	0.38	96%	<0.001
1000–2000	10	0.31	0.21	0.42	98%	
>2000	13	0.27	0.23	0.31	99%	
Collection date (k = 28)						
January-June 2020	17	0.23	0.17	0.30	99%	<0.001
July-December 2020	5	0.32	0.19	0.44	98%	
2021–2022	6	0.37	0.29	0.46	94%	
Methodological quality (k = 31)						
Poor	10	0.30	0.22	0.37	96%	<0.001
Fair	9	0.29	0.20	0.39	97%	
Good	12	0.27	0.19	0.35	99%	
Depression measures (k = 31)						
PHQ	17	0.23	0.18	<0.001	99%	<0.001
CES	4	0.37	0.17	0.56	99%	
Other	10	0.35	0.28	0.43	78%	
Social support measures (k = 31)						
MSPSS	19	0.29	0.25	<0.001	96%	<0.001
Other	12	0.27	0.18	0.37	99%	
**DV = Social support; IV = Anxiety**						
**Covariate**	**K**	**r**	**Lower CI**	**Upper CI**	**Heterogeneity**	***p*-value**
Proportion of females (k = 28)						
<50%	5	0.33	0.22	0.44	97%	<0.001
50 to 70%	15	0.17	0.11	0.23	98%	
>70%	8	0.14	0.08	0.21	94%	
Age groups (k = 26)						
<30 years	16	0.23	0.16	<0.001	99%	<0.001
30 to 40	7	0.18	0.09	0.28	92%	
>40 years	5	0.12	0.05	0.19	93%	
Region (k = 28)						
China	11	0.22	0.17	<0.001	99%	<0.001
Europe	7	0.13	0.06	0.21	93%	
Developed	7	0.18	0.05	0.32	98%	
Developing	3	0.23	0.08	0.38	96%	
Stringency index (k = 27)						
<70	5	0.21	0.12	0.31	95%	<0.001
70–80	14	0.20	0.13	0.27	99%	
>80	8	0.14	0.07	0.22	98%	
Sample size (k = 28)						
<1000	8	0.18	0.09	0.27	93%	<0.001
1000–2000	8	0.25	0.13	0.36	97%	
>2000	12	0.17	0.11	0.22	99%	
Collection date (k = 26)						
January-June 2020	21	0.16	0.11	0.22	98%	<0.001
July-December 2020	3	0.27	0.17	0.38	97%	
2021–2022	2	0.29	0.15	0.43	91%	
Methodological quality (k = 28)						
Poor	8	0.18	0.10	0.26	98%	<0.001
Fair	10	0.23	0.14	0.32	97%	
Good	10	0.16	0.09	0.24	98%	
Anxiety measures (k = 28)						
GAD	16	0.15	0.09	<0.001	98%	<0.001
DASS	5	0.21	0.10	0.32	94%	
Other	7	0.28	0.21	0.35	89%	
Social support measures (k = 28)						
MSPSS	19	0.19	0.14	<0.001	98%	<0.001
Other	9	0.20	0.09	0.31	99%	
**DV = Social support; IV = Posttraumatic stress**						
**Covariate**	**K**	**r**	**Lower CI**	**Upper CI**	**Heterogeneity**	***p*-value**
Proportion of females (k = 7)						
<50%	0	-	-	-	-	<0.01
50 to 70%	4	0.22	0.12	0.32	95%	
>70%	3	0.14	0.03	0.26	93%	
Age groups (k = 7)						
<30 years	4	0.18	0.09	<0.01	95%	<0.01
30 to 40	3	0.19	0.04	0.35	97%	
>40 years	0	-	-	-	-	
Region (k = 7)						
China	3	0.23	0.10		96%	<0.01
Europe	1	0.08	0.05	<0.01	-	
Developed	3	0.17	0.06	0.27	88%	
Developing	0	-	-	-	-	
Stringency index (k = 6)						
<70	0	-	-	-	-	-
70–80	6	0.17	0.09	0.26	97%	
>80	0	-	-	-	-	
Sample size (k = 7)						
<1000	2	0.11	0.05	0.18	36%	<0.01
1000–2000	3	0.24	0.10	0.37	96%	
>2000	2	0.17	0.00	0.33	99%	
Collection date (k = 7)						
January-June 2020	6	0.17	0.09	0.26	97%	<0.01
July-December 2020	0	-	-	-	-	
2021–2022	1	0.26	0.20	0.31	-	
Methodological quality (k = 7)						
Poor	1	0.08	0.05	0.12	-	<0.01
Fair	3	0.15	0.05	0.26	91%	
Good	3	0.25	0.16	0.35	89%	
PTSS measures (k = 7)						
PCL5	2	0.25	0.07	<0.01	92%	<0.01
Other	5	0.16	0.08	0.24	98%	
Social support measures (k = 7)						
MSPSS	7	0.18	0.11	<0.01	97%	<0.01
Other	0	-	-	-	-	

NOTE: Some studies were excluded from the subgroup analysis due to missing values.

## Data Availability

Data will be made available upon request made to the corresponding author.

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
