# Peer review of "The Association of Social Support and Loneliness with Symptoms of Depression, Anxiety, and Posttraumatic Stress during the COVID-19 Pandemic: A Meta-Analysis"

_ijerph, 2023, doi:10.3390/ijerph20042765_

Round 1

Reviewer 1 Report

I am not an expert of systematic review and meta-analysis, so I have evaluated the manuscript based on my overall view on how the manuscript is formulated. To my understanding the researchers have carefully followed the guidelines provided in the PRISMA statement. The review protocol has also been registered in PROSPERO, which is also recommended in systematic reviews.

The topic of the paper is interesting and relevant across disciplines. As the authors state, it is important to gain more knowledge on which aspects of social wellbeing have had the most impact on mental health during the Covid-19 pandemic. The authors have made a thorough investigation to gain a systematic picture of what is known of these associations. Limitations of the study, such as focusing only on cross-sectional studies, are also discussed.

I have only a few minor suggestions:

-        On page 2, line 71 “vulnerable groups” are mentioned but it is unclear what groups are here considered to me vulnerable. Some examples would be important here as "vulnerability" is an ambiguous concept.

-        On page 3, line 109 word “elderly” should be changed to “older adults” as "elderly" is broadly considered as a ageist concept and unsuitable for describing people over 60.

-        On page 20, line 351 “vulnerable regions” are mentioned. I would recommend using the term "disadvantaged regions" and giving also here some concrete examples on what is meant by this.

-        Conclusions section focuses on highlighting the importance of loneliness interventions. It is not very clear what psychological assistance means in the context of loneliness. Here it would be useful to refer to existing knowledge on loneliness interventions.

Author Response

Thank you for the opportunity to improve the manuscript entitled “The association of social support and loneliness with symptoms of depression, anxiety, and posttraumatic stress during the COVID-19 pandemic: a meta-analysis”. We would like to express our sincere gratitude for the suggested corrections. We have addressed all issues raised by the reviewers’ and all changes to the manuscript are highlighted with Word Track Changes.

Response to Reviewer 1 Comments

-  On page 2, line 71 “vulnerable groups” are mentioned but it is unclear what groups are here considered to me vulnerable. Some examples would be important here as "vulnerability" is an ambiguous concept.

Reply: Thank you for your suggestion. We have added some examples to clarify what groups are considered vulnerable.

 “Post-traumatic stress symptoms (PTSS), as already observed during and after the SARS pandemic of 2003, are likely to appear and increase in the long-term following the COVID-19 pandemic, particularly among the most vulnerable groups (e.g., COVID-19 patients and their close contacts, health care workers and other hospital staff, persons with a psychiatric illness history or with underlying health conditions, older people, individuals who reside in high COVID-19 prevalence areas, etc.)  [27–29]. ” – lines 72 - 75, page 2

-  On page 3, line 109 word “elderly” should be changed to “older adults” as "elderly" is broadly considered as a ageist concept and unsuitable for describing people over 60.

Reply: Thank you for raising this issue. We have changed it accordingly.

 “We also excluded studies focused on older adults (>60 years)… ”  – line 112, page 3

-  On page 20, line 351 “vulnerable regions” are mentioned. I would recommend using the term "disadvantaged regions" and giving also here some concrete examples on what is meant by this.

Reply: Thank you for your comment. Following your recommendation, we have changed the term and defined it in more detail.

“Our results are also in line with previous evidence suggesting that the beneficial effects of social support on mental health are stronger in the most deprived regions and neighbourhoods (i.e., with lower socioeconomic conditions and social capital and higher poverty), where inhabitants would be more likely to establish reciprocity networks with neighbours due to the absence of other resources [137-139].”  – lines 354 – 358, page 20

-  Conclusions section focuses on highlighting the importance of loneliness interventions. It is not very clear what psychological assistance means in the context of loneliness. Here it would be useful to refer to existing knowledge on loneliness interventions.

Reply: Thank you for your comment. Following your suggestion, we have clarified the meaning of psychological interventions to target loneliness.

“These interventions directed towards people feeling loneliness should aim (i) to provide psychological assistance promoting changes in their social behaviour (i.e., targeting their maladaptive social perception and cognitive biases towards loneliness [145,146]) and (ii) to increase their chances of establishing satisfactory social contacts, considering the target population and the effects of moderator variables such as gender, setting, and age.”– lines 411 - 414, page 21

Reviewer 2 Report

I congratulate the authors on an interesting study and want to make only two small points. My first comment refers to the introduction. The authors very briefly describe the relationship between loneliness and social support, although this relationship is well covered in a large number of studies. Perhaps the authors can expand the introduction, and add these or some other sources:

Clifton, K., Gao, F., Jabbari, J., Van Aman, M., Dulle, P., Hanson, J., & Wildes, T. M. (2022). Loneliness, social isolation, and social support in older adults with active cancer during the COVID-19 pandemic. Journal of Geriatric Oncology, 13(8), 1122–1131. https://doi.org/10.1016/j.jgo.2022.08.003

Kan, Z., Søegaard, E. G. I., Siqveland, J., Hussain, A., Hanssen-Bauer, K., Jensen, P., Heiervang, K. S., Ringen, P. A., Ekeberg, Ø., Hem, E., Heir, T., & Thapa, S. B. (2022). Coping, social support and loneliness during the COVID-19 pandemic and their effect on depression and anxiety: Patients’ experiences in community mental health centers in Norway. Healthcare, 10(5), 875. https://doi.org/10.3390/healthcare10050875

Pineda, C. N., Naz, M. P., Ortiz, A., Ouano, E. L., Padua, N. P., Paronable, J., Pelayo, J. M., Regalado, M. C., & Torres, G. C. S. (2022). Resilience, social support, loneliness and quality of life during COVID-19 pandemic: A structural equation model. Nurse Education in Practice, 64, 103419. https://doi.org/10.1016/j.nepr.2022.103419

Saltzman, L. Y., Hansel, T. C., & Bordnick, P. S. (2020). Loneliness, isolation, and social support factors in post-COVID-19 mental health. Psychological Trauma: Theory, Research, Practice, and Policy, 12(S1), S55–S57. https://doi.org/10.1037/tra0000703

My second small comment is editorial in nature: on line 67 the authors forgot to remove the initial in the reference to the source (J. Wang et al., 2018).

Author Response

Thank you for the opportunity to improve the manuscript entitled “The association of social support and loneliness with symptoms of depression, anxiety, and posttraumatic stress during the COVID-19 pandemic: a meta-analysis”. We would like to express our sincere gratitude for the suggested corrections. We have addressed all issues raised by the reviewers’ and all changes to the manuscript are highlighted with Word Track Changes.

Response to Reviewer 2 Comments

I congratulate the authors on an interesting study and want to make only two small points. My first comment refers to the introduction. The authors very briefly describe the relationship between loneliness and social support, although this relationship is well covered in a large number of studies. Perhaps the authors can expand the introduction, and add these or some other sources:

  • Clifton, K., Gao, F., Jabbari, J., Van Aman, M., Dulle, P., Hanson, J., & Wildes, T. M. (2022). Loneliness, social isolation, and social support in older adults with active cancer during the COVID-19 pandemic. Journal of Geriatric Oncology, 13(8), 1122–1131. https://doi.org/10.1016/j.jgo.2022.08.003
  • Kan, Z., Søegaard, E. G. I., Siqveland, J., Hussain, A., Hanssen-Bauer, K., Jensen, P., Heiervang, K. S., Ringen, P. A., Ekeberg, Ø., Hem, E., Heir, T., & Thapa, S. B. (2022). Coping, social support and loneliness during the COVID-19 pandemic and their effect on depression and anxiety: Patients’ experiences in community mental health centers in Norway. Healthcare, 10(5), 875. https://doi.org/10.3390/healthcare10050875
  • Pineda, C. N., Naz, M. P., Ortiz, A., Ouano, E. L., Padua, N. P., Paronable, J., Pelayo, J. M., Regalado, M. C., & Torres, G. C. S. (2022). Resilience, social support, loneliness and quality of life during COVID-19 pandemic: A structural equation model. Nurse Education in Practice, 64, 103419. https://doi.org/10.1016/j.nepr.2022.103419
  • Saltzman, L. Y., Hansel, T. C., & Bordnick, P. S. (2020). Loneliness, isolation, and social support factors in post-COVID-19 mental health. Psychological Trauma: Theory, Research, Practice, and Policy, 12(S1), S55–S57. https://doi.org/10.1037/tra0000703

Reply: Thank you for your comment. Following your suggestion, we have expanded the description of the relationship between loneliness and social support in the “Introduction” section adding some of the recommended references. Furthermore, we have further discussed in the “Discussion” section this relationship and the similarities and differences between these two constructs.

Introduction section:

“According to previous research, loneliness and low social support are among the social determinants most closely related to mental health compared to socioeconomic, material, and behavioural factors [13,17]. These relationships mainly occur with loneliness as the origin [18].  Pre-pandemic and during the pandemic investigations indicate that low social support boosts the development of loneliness [19–22] and that the effect of social support on mental health is mediated by loneliness [23–25]. Both factors increase the odds of having symptoms of depression and anxiety [18,26].”– lines 65 - 68, page 2

Discussion section:

“The smaller effect size obtained in those associations where social support was the independent variable could be explained by (i) the fact that loneliness might mediate the relationship between social support and mental health [23,24], and (ii) the substantial overlap between these two constructs and the instruments used to measure them. Loneliness and social support are both strictly linked with an individual’s social system and are interconnected concepts that affect one another. They partially share some conceptual aspects but are distinguished by the theoretical interpretation and definition of the concepts, the individual’s experience, perceptions, and social exchanges, and its public connotations [133,134].”  – lines 335 - 339, page 20

My second small comment is editorial in nature: on line 67 the authors forgot to remove the initial in the reference to the source (J. Wang et al., 2018).

Reply: Thank you for your observation. We have updated all the references according to the journal’s reference style, so now the in-text references are numbered. We will keep this observation in mind for future works.

Reviewer 3 Report

Dear Authors,

The manuscript is considered to be well planned and developed.

Whenever a systematic review is carried out, other databases and other search limits could be established, but this is a decision that is understood to be consensual and justified by the research team. 

Three minor observations are made in section 2.1. Eligibility criteria:

- line 97: specify that IV is an abbreviation of independent variables by putting in brackets.

- line 98: specify that DV is an abbreviation for dependent variables by putting in parentheses

- line 103: since it has been specified which types of studies have been included, it is understood that the rest are excluded.

On the other hand, the search strategies could not be evaluated since Appendix A is not available to the reviewer in the documentation.

Author Response

Thank you for the opportunity to improve the manuscript entitled “The association of social support and loneliness with symptoms of depression, anxiety, and posttraumatic stress during the COVID-19 pandemic: a meta-analysis”. We would like to express our sincere gratitude for the suggested corrections. We have addressed all issues raised by the reviewers’ and all changes to the manuscript are highlighted with Word Track Changes.

Response to Reviewer 3 Comments

The manuscript is considered to be well planned and developed.

Whenever a systematic review is carried out, other databases and other search limits could be established, but this is a decision that is understood to be consensual and justified by the research team.

Three minor observations are made in section 2.1. Eligibility criteria:

  • line 97: specify that IV is an abbreviation of independent variables by putting in brackets.
  • line 98: specify that DV is an abbreviation for dependent variables by putting in parentheses

Reply: Thank you for your suggestions. We have specified it as recommended.

“…the associations of social support or loneliness (i.e., independent variables [IV]) with symptoms of depression, anxiety, or posttraumatic stress (PTS) (i.e., dependent variables [DV]).” – lines 100 - 101, page 3

  • line 103: since it has been specified which types of studies have been included, it is understood that the rest are excluded.

Reply: Thank you for your comment. We have erased the sentence as it was redundant.

Reviewer 4 Report

Dear Authors,

It is a well conducted meta-analysis following standardized methodology with clear and specific study question, specific eligibility/exclusion criteria for included studies, appropriate/comprehensive search strategy with the aim of including all relevant studies, quality assessment of individual studies with standardized tools, data abstraction, using appropriate mathematical modeling & statistical techniques, evaluating the results, assessment of heterogeneity & attempt to explain the heterogeneity, assessing for publication bias and appropriate conclusion. However, inclusion of poor-quality studies (22 out of 73) can render the summary estimate invalid, introducing a significant bias. This issue should be dealt with during the design phase. Please state the reason you have chosen to include such studies. Also, the high observed heterogeneity should be interpreted with caution, although you attempted to offer explanations by investigating sources of heterogeneity and identify several moderator variables; but apart from sociodemographic and/or methodological differences between studies there might be other unidentified moderator variables and/or there is always the possibility of true variability in outcomes.

Minor recommendations:

1. Final paragraph in discussion section, lines 340-371 is too long. I suggest that you divide this paragraph in at least 3 shorter paragraphs, dealing with separate moderators, to make the text reader-friendly.

2. Please change the reference style in the text according to the journal’s guidelines; from authors names to numbers and check the corresponding references again (some of them seem to be out of place).   

Author Response

Thank you for the opportunity to improve the manuscript entitled “The association of social support and loneliness with symptoms of depression, anxiety, and posttraumatic stress during the COVID-19 pandemic: a meta-analysis”. We would like to express our sincere gratitude for the suggested corrections. We have addressed all issues raised by the reviewers’ and all changes to the manuscript are highlighted with Word Track Changes.

Response to Reviewer 4 Comments

It is a well conducted meta-analysis following standardized methodology with clear and specific study question, specific eligibility/exclusion criteria for included studies, appropriate/comprehensive search strategy with the aim of including all relevant studies, quality assessment of individual studies with standardized tools, data abstraction, using appropriate mathematical modeling & statistical techniques, evaluating the results, assessment of heterogeneity & attempt to explain the heterogeneity, assessing for publication bias and appropriate conclusion. However, inclusion of poor-quality studies (22 out of 73) can render the summary estimate invalid, introducing a significant bias. This issue should be dealt with during the design phase. Please state the reason you have chosen to include such studies.

Reply: Thank you for your comment. The reason why we included studies with poor methodological quality was because they met our inclusion criteria. Methodological quality was not predefined in our study protocol as an exclusion criteria. Nevertheless, as you pointed out, 22 out of 73 is a considerable number of studies, reason why we included the quality of the studies in the subgroup analysis to assess the potential effect of poor quality studies in the overall estimates. In ‘Table 2’ and ‘Table 3’ we can see the correlation estimates with just the good and high quality studies, and we can observe that low-quality studies did not have a significant effect on our overall results.

Following your comment, we have reported this issue in the “4.1. Strengths & limitations” section of our manuscript.

“Third, although 22 out of the 73 included studies had a poor methodological quality, the exclusion of the poor-quality studies did not have an impact on our overall results, as it can be seen in the subgroup analyses.”  – line 389 – 392, page 21

Also, the high observed heterogeneity should be interpreted with caution, although you attempted to offer explanations by investigating sources of heterogeneity and identify several moderator variables; but apart from sociodemographic and/or methodological differences between studies there might be other unidentified moderator variables and/or there is always the possibility of true variability in outcomes.

Reply: Thank you for your comment. Following your recommendation, we have emphasized this in the “Limitations” section.

“We explored methodological and theoretical factors moderating the correlation of the associations, but it is likely that other factors such as sociodemographic and socioeconomic characteristics of the study samples may have also contributed to the high heterogeneity of the results. Therefore, we should cautiously interpret the findings of the present study and future studies should try to identify further explanatory factors. ” – lines 400 - 402, page 21

Minor recommendations:

  1. Final paragraph in discussion section, lines 340-371 is too long. I suggest that you divide this paragraph in at least 3 shorter paragraphs, dealing with separate moderators, to make the text reader-friendly.

Reply: Thank you for your suggestion, we have divided the paragraph into shorter paragraphs according to the moderators that are discussed.

  1. Please change the reference style in the text according to the journal’s guidelines; from authors names to numbers and check the corresponding references again (some of them seem to be out of place).

Reply: Thank you for your revision. We have updated the references according to the journal’s guidelines and we have also revised the references position within the text.

Reviewer 5 Report

This paper examines the effect of loneliness and social support on depression, anxiety and PTSD in the general population during the COVID-19 pandemic employing a meta-analysis approach. The issue is relevant and timely considering the body of research on the protective role of affiliative links and social interaction on countering psychological distress symptoms during pandemics. Teasing out the specific effect of loneliness and social support as different constructs tapping on how social interactions influence on one’s health and wellbeing social interaction represents an important and useful approach and I commend the authors for such undertaking. The authors synthesize 73 studies considered appropriate for inclusion in the final analysis, corresponding to a total of 1,020,461 participants and found a medium effect size for the association of loneliness with symptoms of depression (r=0.49), anxiety (r=0.40) and PTSD (r=0.38) while a small effect size was found for the association between social support and symptoms of depression (r=0.29), anxiety (r=0.19) and PTS (r=0.18). The paper is well written, methods and results are clearly described.

 However, it is not very clear how these two constructs differ from one another, especially as one considers the heterogeneity of the measures used in the target studies to measure them. For instance, in page 2 (lines 60-64) authors state that social support refers more to the “objective” aspect of social relationships whereas “loneliness” refers to the subjective feeling of an unpleasant gap between the desired and available social relationships both quantitatively and qualitatively.

 It would seem from this definition that there is some overlap (naturally) in terms of these constructs and that “loneliness” might include (at least partially) aspects relevant to the construct of “social support”. A close examination of the measures used in the studies included in the analysis would tend to confirm this idea (as for instance is the case of the De Jong Gierfield scale, or the Italians Social and emotional loneliness which are not unidimensional scales).

 It follows that the difference in the effect sizes found for the two constructs, that of loneliness being higher than that of social support, might perhaps be attributed to the fact that the construct of “loneliness” as measured by instruments used in the target studies partially overlaps and includes aspects referred to as “social support”?.

 I believe it would greatly enhance the quality of the current paper if the authors briefly discussed on the potential of construct overlap by examining overlaps in the instruments used in the included studies. Perhaps this point could also be added in the discussion section as a potential factor influencing current results.

Author Response

Thank you for the opportunity to improve the manuscript entitled “The association of social support and loneliness with symptoms of depression, anxiety, and posttraumatic stress during the COVID-19 pandemic: a meta-analysis”. We would like to express our sincere gratitude for the suggested corrections. We have addressed all issues raised by the reviewers’ and all changes to the manuscript are highlighted with Word Track Changes.

Response to Reviewer 5 Comments

However, it is not very clear how these two constructs differ from one another, especially as one considers the heterogeneity of the measures used in the target studies to measure them. For instance, in page 2 (lines 60-64) authors state that social support refers more to the “objective” aspect of social relationships whereas “loneliness” refers to the subjective feeling of an unpleasant gap between the desired and available social relationships both quantitatively and qualitatively. It would seem from this definition that there is some overlap (naturally) in terms of these constructs and that “loneliness” might include (at least partially) aspects relevant to the construct of “social support”. A close examination of the measures used in the studies included in the analysis would tend to confirm this idea (as for instance is the case of the De Jong Gierfield scale, or the Italians Social and emotional loneliness which are not unidimensional scales). It follows that the difference in the effect sizes found for the two constructs, that of loneliness being higher than that of social support, might perhaps be attributed to the fact that the construct of “loneliness” as measured by instruments used in the target studies partially overlaps and includes aspects referred to as “social support”?. I believe it would greatly enhance the quality of the current paper if the authors briefly discussed on the potential of construct overlap by examining overlaps in the instruments used in the included studies. Perhaps this point could also be added in the discussion section as a potential factor influencing current results.

Reply: Thank you for your comment. Following your suggestion, we have expanded the description of the relationship between loneliness and social support in the “Introduction” section and we have further discussed in the “Discussion” section this relationship and the possible overlap between these two constructs and the instruments used to measure them.

Introduction section:

“According to previous research, loneliness and low social support are among the social determinants most closely related to mental health compared to socioeconomic, material, and behavioural factors [13,17]. These relationships mainly occur with loneliness as the origin [18].  Pre-pandemic and during the pandemic investigations indicate that low social support boosts the development of loneliness [19–22] and that the effect of social support on mental health is mediated by loneliness [23–25]. Both factors increase the odds of having symptoms of depression and anxiety [18,26].”– lines 65 - 68, page 2

Discussion section:

“The smaller effect size obtained in those associations where social support was the independent variable could be explained by (i) the fact that loneliness might mediate the relationship between social support and mental health [23,24], and (ii) the substantial overlap between these two constructs and the instruments to measure them. Loneliness and social support are both strictly linked with an individual’s social system and are interconnected concepts that affect one another. They partially share some conceptual aspects but are distinguished by the theoretical interpretation and definition of the concepts, the individual’s experience, perceptions, and social exchanges, and its public connotations [133,134].”  – lines 335 - 339, page 20
